

# First record of *Trichinella* in *Leopardus guigna* (Carnivora, Felidae) and *Galictis cuja* (Carnivora, Mustelidae): new hosts in Chile

Diana Maritza Echeverry[1], AnaLía Henríquez[2], Pablo Oyarzún-Ruiz[1], Maria Carolina Silva-de la Fuente[3], Rene Ortega[1], Daniel Sandoval[1] and Carlos Landaeta-Aqueveque[1]

[1] Facultad de Ciencias Veterinarias, Universidad de Concepción, Chillán, Región de Biobío/Ñuble, Chile
[2] Facultad de Medicina Veterinaria, Universidad San Sebastián, Concepción, Biobío, Chile
[3] Facultad de Ciencias Veterinarias, Universidad Austral de Chile, Valdivia, Los Ríos, Chile

Corresponding author
Carlos Landaeta-Aqueveque,
clandaeta@udec.cl

## ABSTRACT

**Background**. Trichinellosis is a zoonotic disease with a worldwide distribution. It is caused by several species of nematodes in the genus *Trichinella. Trichinella* spp. are transmitted through predation or carrion consumption and occur in domestic and sylvatic cycles. In humans trichinellosis occurs due to the consumption of raw or undercooked, infected meat and is mainly associated with the household slaughter of pigs or the consumption of game animals without veterinary inspection, a cultural practice that is difficult to resolve. Therefore, knowledge of this parasite's reservoir is relevant for better implementing public health strategies. The aim of this study was to assess the presence of *Trichinella* sp. in several carnivore and omnivore vertebrates in central-southern Chile.

**Methods**. We collected muscle tissue from a total of 53 animals from 15 species and were digested to detect *Trichinella* larvae which were further identified to species level using molecular techniques.

**Results**. We detected *Trichinella* larvae in *Leopardus guigna* (Felidae) and *Galictis cuja* (Mustelidae). We identified the larvae collected from *L. guigna* as *Trichinella spiralis*, but we were unable to molecularly characterize the larvae from *G. cuja*. This is the first record of *Trichinella* in a native mustelid of South America and the first record of *T. spiralis* in *L. guigna*. This study identified two novel hosts; however, further work is needed to identify the role that these and other hosts play in the cycle of *Trichinella* in Chile.

## INTRODUCTION

Trichinellosis is a disease that is distributed worldwide and is caused by nematodes in the genus *Trichinella* (*Korhonen et al., 2016*). It is considered neglected and emerging in some regions (*Dupouy-Camet, 1999*; *Murrell & Pozio, 2000*; *Bruschi, 2012*; *Boutsini et al., 2014*).

*Trichinella* nematodes are transmitted from animals to humans by the ingestion of raw or undercooked infected meat.

*Trichinella* is transmitted among non-human animals via predation and carrion consumption; therefore, it circulates among carnivorous and omnivorous vertebrates. Two cycles have been described: the domestic (encompassing mainly pigs, rats, dogs, and cats) and the sylvatic (encompassing free-range vertebrates) cycles (*Pozio, 2000*; *Pozio, 2007*; *Loutfy et al., 1999*). These cycles can be connected and fed back by invasive rats and other synanthropic animals (*Pozio, 2000*). The domestic cycle was the primary cause of human infections; however, improvements in pork production have reduced outbreaks globally (*Devleesschauwer et al., 2015*; *Murrell, 2016*). The improvements to pork production changed the epidemiology of trichinellosis in human populations. *Trichinella* infections now primarily occur during the consumption of meat from unregulated sources, mainly backyard pork production and the consumption of game animals (*Pozio, 2014*; *Tryland et al., 2014*; *Fichi et al., 2015*; *Kärssin et al., 2017*).

At present, there are 10 recognized species of *Trichinella* around the world and three additional genotypes that have not yet been identified as distinct species (*Korhonen et al., 2016*; *Sharma et al., 2020*). Most species infect only mammals (*Klun et al., 2019*; *Bilska-Zając et al., 2020*), including marine mammals (*Tryland et al., 2014*; *Pasqualetti et al., 2018*). However, *Trichinella pseudospiralis* Garkavi, 1972 also infects birds, and *Trichinella zimbabwensis* Pozio et al., 2002 and *Trichinella papuae* Pozio et al., 1999 infect reptile hosts (*Korhonen et al., 2016*). Thus, obtaining ecological and epidemiological knowledge of the transmission cycle is relevant for reducing the incidence of this parasite.

In South America, *Trichinella* spp. infections have been detected in Argentina, Bolivia, Chile (larvae isolation), Brazil, and Ecuador (antibody detection) with most studies focusing on the domestic cycle (*Bjorland et al., 1993*; *Ribicich et al., 2020*). Four species have been reported: *Trichinella spiralis* Owen, 1835, *Trichinella patagoniensis* Krivokapich et al., 2012, *Trichinella britovi* Pozio et al., 1992, and *T. pseudospiralis* (*Krivokapich et al., 2006*; *Krivokapich et al., 2012*; *Krivokapich et al., 2015*; *Krivokapich et al., 2019*). Additionally, *Trichinella* infections have been documented from eight wild species: cougar (*Puma concolor* Linnaeus, 1771), wild boar (*Sus scrofa* Linnaeus, 1758), fox (*Lycalopex gymnocercus gracilis* Fischer, 1814), opossum (*Didelphis albiventris* Lund, 1840), sea lion (*Otaria flavescens* Shaw, 1800), pecarí (*Tayassu tajacu* Palmer, 1897), armadillo (*Chaetophractus villosus* Desmerest, 1804), and pericote (*Graomys centralis* Thomas, 1902) (*Minoprio, Abdon & Abdon, 1967*; *Ribicich et al., 2020*; *Soria et al., 2010*).

In Chile, the domestic cycle is fairly well-studied (*Alcaíno & Arenas, 1981*; *Schenone et al., 2002*; *Landaeta-Aqueveque et al., 2021*), but the sylvatic cycle is largely unknown. *Trichinella spiralis* is the sole species that has been reported in Chile (*Schenone et al., 2002*; *Landaeta-Aqueveque et al., 2015*; *Hidalgo et al., 2019*; *Echeverry et al., 2021*; *Espinoza-Rojas et al., 2021*). Among non-domestic animals, cougars, American minks (*Neovison vison* Schreber, 1777) and wild boar are the only wild/feral hosts with documented infections (*Landaeta-Aqueveque et al., 2015*; *Hidalgo et al., 2019*; *Echeverry et al., 2021*; *Espinoza-Rojas et al., 2021*). In addition to those reports, other studies have not found infected animals (*Alvarez et al., 1970*; *González-Acuña et al., 2010*; *Ramirez-Pizarro et al., 2019*). Therefore,

 

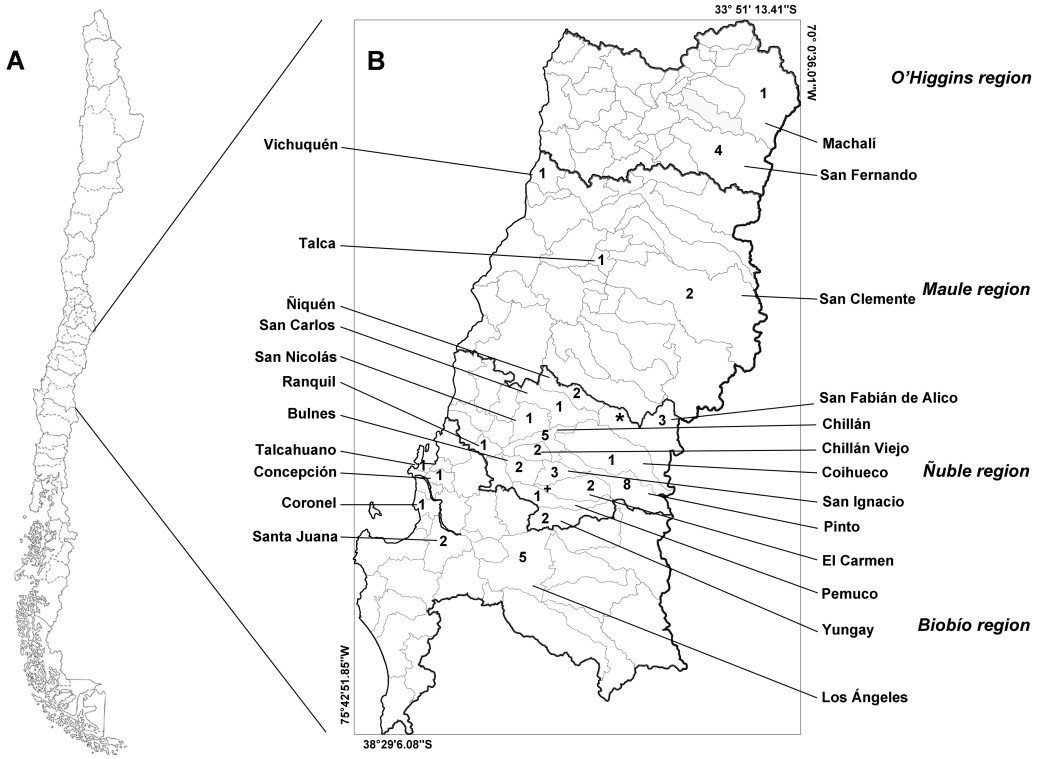

**Figure 1** **Map of Chile (A) and the studied administrative regions (B).** The italicized text indicates the name of the regions, and the Roman text indicates the name of the communes. Infected animals are presented with the symbols ''+'' (*Leopardus guigna*) and ''*'' (*Galictis cuja*). The numbers indicate the number of animals examined in each commune. Thick lines indicate the regional limits, while thin lines indicate the limits of the communes.

the objective of this study was to assess the presence of *Trichinella* sp. in carnivorous and omnivorous wild vertebrates from south-central Chile.

## MATERIALS & METHODS

The study area includes four administrative regions of Chile: the O'Higgins, Maule, Ñuble, and Biobío regions (Fig. 1). These regions feature a transitional climate that falls somewhere between the classifications of warm Mediterranean (Csb, after Köpen classification) and wet temperate oceanic (Cfb, after Köpen classification). These regions lie within the limits between central and southern Chile.

This study considered animals that were found dead, mainly run over by a vehicle, or that died in wild animal rescue/rehabilitation centers (Fauna Rehabilitation Center of the Universidad de Concepción; Wild Fauna Rehabilitation Center of the Universidad San Sebastián) from 2013 to 2020. We examined at least 1 g of muscle (10 g, when possible) of these animals to determine the presence of *Trichinella* spp. larvae. We then selected the following muscles for parasitological examination: the diaphragm, masseter, tongue, quadriceps (in mammals), pectoral (in birds), and intercostals (in all animals).

We performed artificial digestion of the muscles following the method described by *Gajadhar et al. (2019)* and preserved the larvae in 96% ethanol. For molecular identification, we extracted DNA from a pool of 10 *Trichinella* larvae isolated from each positive animal using the DNeasy Blood & Tissue Kit (Qiagen, Hilden, Germany) and used 10 ng of DNA for identification at the species level by nested polymerase chain reaction (PCR), following a modification of the protocol of *Zarlenga et al. (1999)*. We performed the reactions at a final volume of 25 μL. We used the following primers: *Ne* forward (5′-TCTTGGTGGTAGTAGC-3′) and reverse (5′-GCGATTGAGTTGAACGC-3′) in the first PCR (0.5 μM of each primer), and 12.5 μL of GoTaq Green Master Mix (Promega Corporation, Madison, WI, USA). We amplified the DNA in a thermocycler (MultiGene™ OptiMax Thermal Cycler; Labnet International, Inc., Edison, NJ, USA) under the following cycling conditions: 95 °C ×1 min for initial denaturation, followed by 40 cycles of 95 °C ×30 s; 56 °C ×1 min, and 72 °C ×1 min; and a final extension of 72 °C ×2 min. Then, we used 0.5 μM of each Primers *I* forward (5′-GTTCCATGTGAACAGCAG-3′) and reverse (5′- CGAAAACATACGACAACTGC-3′) in a second PCR under same conditions with an annealing temperature of 55 °C. The PCR products were subjected to electrophoresis in 2% agarose gel. We used master mix without the DNA as the negative control, and *T. spiralis* larvae obtained from a previous study (*Landaeta-Aqueveque et al., 2015*) as a positive control of the PCR.

Bioethical considerations: this study met the International Guiding Principles for Biomedical Research Involving Animals. The Comité de Ética of the Facultad de Ciencias Veterinarias of the Universidad de Concepción approved the study (CBE-47-2017).

## RESULTS

We collected samples from 53 animals. The sample was composed of 28 mammals, 24 birds and one reptile (Table 1). The weight of the examined muscle samples were at least 10 g with the exception of *D. bozinovici* and *P. chamissonis* with samples sizes of 3 g and 1 g, respectively. *Trichinella* larvae were isolated only from one *Leopardus guigna* Molina, 1782 (güiña; 52 larvae per gram of muscle) and one *Galictis cuja* Molina, 1782 (lesser grison; 0.3 larvae per gram of muscle), both from the Ñuble region (Fig. 1). We were unable to amplify DNA from the larvae isolated from the grison. However, we were able to amplify a PCR product of 173 bp from the güiña which is consistent with our *T. spiralis* positive control (Fig. 2) and the size described for this species (*Pozio & Zarlenga, 2019*).

## DISCUSSION

Detecting *Trichinella* infection is a challenge in wild fauna of Chile because most carnivore vertebrates are protected by law (*SAG, 2012*). This protection is due to conservation concerns or because these animals aid in pest control. Therefore, only invasive animals can be hunted to assess *Trichinella* infection (*Hidalgo et al., 2019*; *Ramirez-Pizarro et al., 2019*; *Espinoza-Rojas et al., 2021*). This has resulted in few studies that have assessed the presence of *Trichinella* infection in native wildlife in Chile (*Alvarez et al., 1970*; *González-Acuña et al., 2010*; *Hidalgo et al., 2013*; *Landaeta-Aqueveque et al., 2015*; *Echeverry et al., 2021*).

**Table 1  Details of examined animals.**

| Species | Infected/Analyzed (%) | Class |
|---|---|---|
| *Glaucidium nana* King, 1828 (Austral pygmy owl) | 0/1 (0) | Aves |
| *Bubo magellanicus* Gmelin, 1788 (Magellanic horned owl) | 0/2 (0) | Aves |
| *Tyto furcata* Temminck, 1827 (American barn owl) | 0/5 (0) | Aves |
| *Strix rufipes* King, 1828 (Rufous-legged owl) | 0/2 (0) | Aves |
| *Parabuteo unicinctus* Temminck, 1824 (Harris' hawk) | 0/11 (0) | Aves |
| *Coragyps atratus* Bechstein, 1793 (Black vulture) | 0/1 (0) | Aves |
| *Cathartes aura* Linnaeus, 1758 (Turkey vulture) | 0/1 (0) | Aves |
| *Pelecanus thagus* Molina, 1782 (Peruvian pelican) | 0/1 (0) | Aves |
| *Grampus griseus* Cuvier, 1812 (Risso's Dolphin) | 0/1 (0) | Mammalia |
| *Otaria flavescens* Shaw, 1800 (South American sealion) | 0/1 (0) | Mammalia |
| *Leopardus guigna* Molina, 1782 (Güiña) | 1/6 (16.67) | Mammalia |
| *Lycalopex culpaeus* Molina, 1782 (Culpeo fox) | 0/2 (0) | Mammalia |
| *Galictis cuja* Molina, 1782 (Lesser grison) | 1/17 (5.88) | Mammalia |
| *Dromiciops bozinovici* D'Elía, Hurtado and D'Anatro, 2016 ('Monito del monte') | 0/1 (0) | Mammalia |
| *Philodryas chamissonis* Wiegmann, 1834 (Long-tailed snake) | 0/1 (0) | Reptilia |

Although one of these studies sampled a broad range of mammalian species including güiñas and lesser grisons, it did not detect *Trichinella* spp. (*Alvarez et al., 1970*).

Studies in Argentina examined another wild felid, the Geoffroy's cat (*Leopardus geoffroyi* D' Orbigny and Gervais, 1844), and the lesser grison with negative results (*Ribicich et al., 2010*; *Winter et al., 2018*). Thus, this is the first record of *Trichinella* spp. larvae in a native mustelid in South America, and the first record of *T. spiralis* in the güiña. The güiña is the second reported South American felid host for this species.

Previously, other mustelids have been reported to host *Trichinella* infections: American mink infected with *T. spiralis* in Chile (*Espinoza-Rojas et al., 2021*) and with *T. spiralis*, *T. britovi*, and *T. pseudospiralis* in Poland (*Hurníková et al., 2016*) and the European badger (*Meles meles* Linnaeus, 1758) infected with *T. britovi* in Romania (*Boros et al., 2020*). Similarly, other felids have reportedly harbored *Trichinella* larvae. *Trichinella* infections have been reported in cougars across most of their range including with *T. spiralis* in Chile (*Landaeta-Aqueveque et al., 2015*; *Echeverry et al., 2021*), *T. patagoniensis* in Argentina (*Krivokapich et al., 2012*), *T. spiralis* and *T. pseudospiralis* in the United States (*Reichard et al., 2015*), *Trichinella nativa* Britov and Boev, 1972, *T. pseudospiralis*, *Trichinella murrelli* Pozio and La Rosa, 2000, and *Trichinella* T6 in Canada (*Gajadhar & Forbes, 2010*). Additionally, infections have been reported in Canadian lynx (*Lynx canadensis* Kerr, 1792) with *Trichinella* T6 in Canada (*Gajadhar & Forbes, 2010*), Eurasian lynx (*Lynx lynx* Schreber, 1777) with *T. britovi*, and the European wildcat (*Felis silvestris* Schreber, 1777) with *T. britovi* and *T. spiralis* (*Pozio et al., 2009*).

The güiña is one of the smallest felids in the world. It is distributed across Chile and Argentina between latitudes of 33°S and 48°S (*Napolitano et al., 2014*). This felid consumes micromammals such as rodents as primary prey (*Delibes-Mateos et al., 2014*; *Figueroa, Corales & Rau, 2018*); consequently, rodents could be the source of infection. Rodents

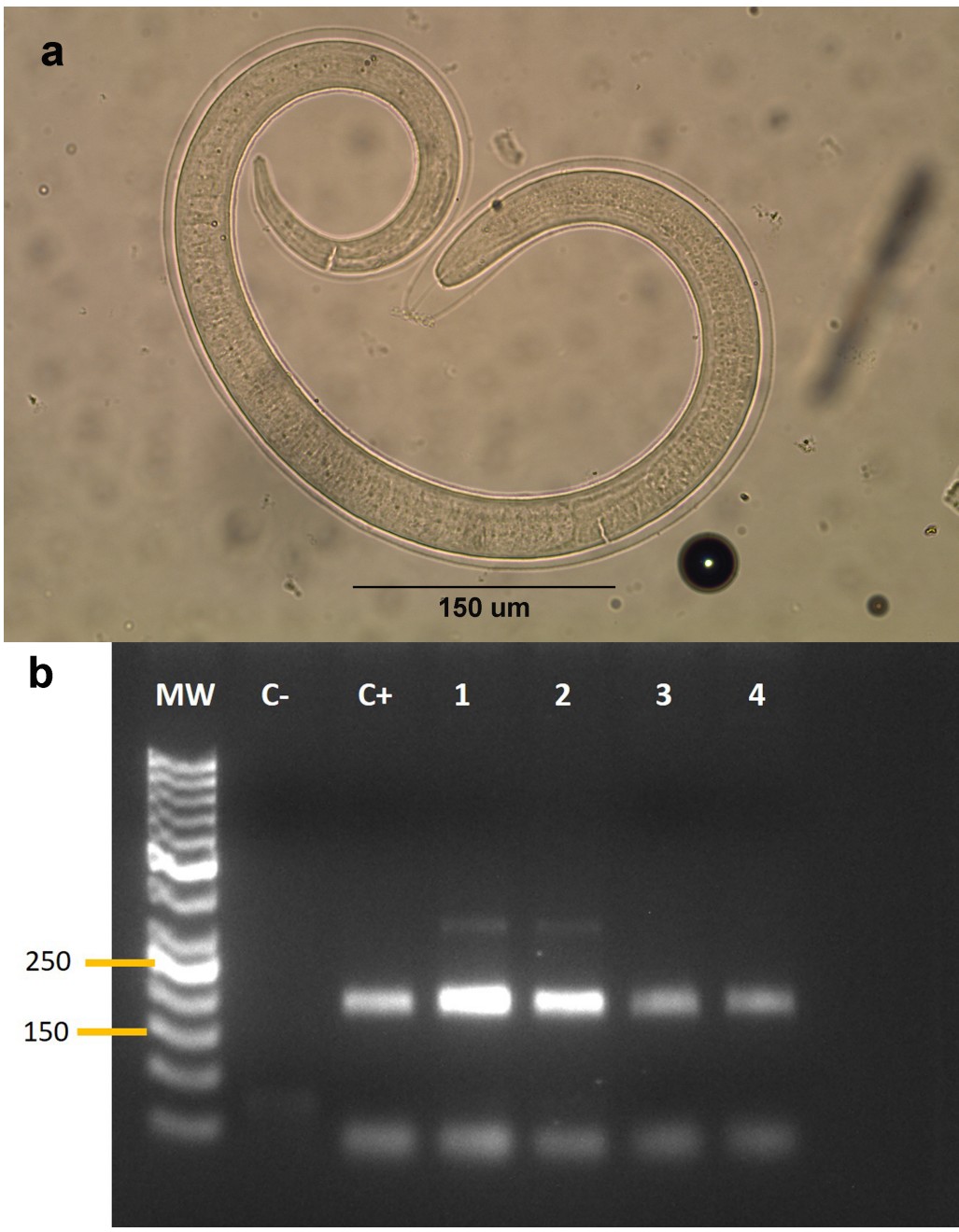

**Figure 2** **(A) Larva of *Trichinella* sp. isolated from a *Galictis cuja*. (B) Gel electrophoresis of PCR products.** (B) MW: Marker of 50 bp. C-: negative control. C+: *Trichinella spiralis* positive control. Lanes 1–4: isolates from *Leopardus guigna*.

have been recognized as hosts of *T. spiralis*, mainly in the domestic environment in Chile (*Schenone et al., 1967*; *Schenone et al., 2002*). This record is in accordance with the fact that güiñas have been frequently infected by pathogens from free-roaming domestic animals (*Ortega et al., 2020*; *Sacristán et al., 2020*); although *T. spiralis* is not an important pathogen

for the health of non-human animals, its presence in the güiña highlights the need for pathogen surveillance in the rural–sylvatic interphase.

The lesser grison is a neotropical mustelid that inhabits an area spanning southern Peru, Uruguay, and Paraguay to southern Chile and Argentina, encompassing several environments (*Prevosti & Travaini, 2005*). It is a generalist predator and rodents comprise an important part of its diet (*Ebensperger, Mella & Simonetti, 1991*; *Zapata et al., 2005*). Given that, and considering how other pathogens have spilled from domestic animals (*Megid et al., 2013*; *Pedrassani et al., 2018*), this species might most likely be infected in domestic environments. However, identification of the *Trichinella* species harbored by the lesser grison helps to better understand the source of infection, given that not all *Trichinella* species identified in South America have been reported in the domestic cycle. For instance, *T. patagoniensis* has been reported only in cougars (*Krivokapich et al., 2008*; *Krivokapich et al., 2012*).

To the best of our knowledge, there are no reports of the güiña as prey of larger predators, whereas the lesser horned owl (*Bubo magellanicus*) is the sole predator to be reported for the lesser grison (*Prevosti & Travaini, 2005*). In that respect, *T. pseudospiralis*, also zoonotic, is the only species of the genus that has reportedly infected birds, and this may be the only species of *Trichinella* that could be transmitted from the grison to the owl. However, this species has not been reported in Chile and one record of a single pig from Argentina represents the only report in South America (*Krivokapich et al., 2015*). Therefore, it is unlikely that this owl could play a role in the sylvatic cycle of *Trichinella* in Chile. Hence, whether güiña and lesser grison participate in the reservoir or constitute dead-end hosts is unknown, and the most likely way for *Trichinella* larvae to be transmitted from these hosts seems to be their consumption by carrion-consuming mammals. Furthermore, human trichinellosis resulting from the direct consumption of a wild mammal has also been reported worldwide (*García et al., 2005*; *Fichi et al., 2015*); however, neither güiñas nor grisons are typical prey for hunters to eat, nor is their hunting permitted by law in Chile (*SAG, 2012*). However, further studies are needed to evaluate these hypotheses.

It is worth noting that the two types of mammal host species reported herein had the largest sample sizes, suggesting that larger samples of other mammals could represent new hosts for *Trichinella*. In contrast, the lack of findings identified by *Alvarez et al. (1970)* may have been due to the real absence of larvae in their samples, as well as to the parasitological technique (trichinoscopy) used, which is of lower sensitivity (*Forbes, Parker & Scandrett, 2003*).

## CONCLUSIONS

This is the first record of *Trichinella* larvae in a native mustelid, *G. cuja*, in South America, as well as the first record of *T. spiralis* in *L. guigna*. Thus, this study increased the number of mammals infected with *Trichinella* larvae in the neotropics, enhancing the need to identify the role played by neotropical animals in the reservoir for humans. This underlies how studying the rural–sylvatic interphase is of utmost importance.

## ACKNOWLEDGEMENTS

In memoriam: The authors dedicate this article to Daniel González-Acuña, who died during the writing of this manuscript prior to submission, and who made significant contributions to this study.

### Funding

This work was supported by the Fondo Nacional de Desarrollo Científico y Tecnológico (ANID/FONDECYT. No. 11170294). The funders had no role in study design, data collection and analysis, decision to publish, or preparation of the manuscript.

### Grant Disclosures

The following grant information was disclosed by the authors:
Fondo Nacional de Desarrollo Científico y Tecnológico (ANID/FONDECYT): 11170294.

### Competing Interests

The authors declare there are no competing interests.

### Author Contributions

- Diana Maritza Echeverry performed the experiments, analyzed the data, prepared figures and/or tables, authored or reviewed drafts of the paper, and approved the final draft.
- AnaLía Henríquez conceived and designed the experiments, authored or reviewed drafts of the paper, contributed with the acquisition of samples, and approved the final draft.
- Pablo Oyarzún-Ruiz performed the experiments, authored or reviewed drafts of the paper, contributed with the acquisition of samples, and approved the final draft.
- Maria Carolina Silva-de la Fuente, Rene Ortega and Daniel Sandoval performed the experiments, authored or reviewed drafts of the paper, and approved the final draft.
- Carlos Landaeta-Aqueveque conceived and designed the experiments, prepared figures and/or tables, authored or reviewed drafts of the paper, obtained the funding, and approved the final draft.

### Animal Ethics

The following information was supplied relating to ethical approvals (i.e., approving body and any reference numbers):

The Comité de Ética of the Facultad de Ciencias Veterinarias of the Universidad de Concepción approved the study (CBE-47-2017).

### Data Availability

The raw numeric data are available in Table 1.

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
