# Peer review of "First record of Trichinella in Leopardus guigna (Carnivora, Felidae) and Galictis cuja (Carnivora, Mustelidae): new hosts in Chile"

_PeerJ, doi:10.7717/peerj.11601_

## Round 0.1 · original submission · Major Revisions

I along with both authors feel the research is valuable and should be published; however, there are major revisions needed before it can be published. Generally, the flow of the writing needs to be improved. Most of this can be accomplished by writing in active voice and I have included an example of the abstract written in active voice at the end of the Editors Comments file.

The molecular methods section also needs to be improved. How did you identify the parasite from the güiña? Was it just the nested PCR and identification of a band? Those primers can amplify multiple Trichinella spp. and further analysis is needed for confirmation of species.

Finally, the results section needs to be improved. You should include more information regarding all the species that were sampled for the study. Knowing where you did not find the parasite is also valuable information. This could be done in a table. Additionally, more molecular information is needed here. A figure of the gel electrophoresis results with appropriate reference samples and/or sequence information.

These changes and a few others are outlined in the EditorsComments.pdf. I look forward to your response.

·

Basic reporting

In some sections of the article, the English language should be improved.

Experimental design

No comment

Validity of the findings

The results contribute to the knowledge of an endemic zoonosis in southern South America, with strong implications for public health.

I recommend publishing the manuscript

Additional comments

Trichinella in wildlife with the first record in a small felid, Leopardus guigna, and a mustelid, Galictis cuja, in South America
The title should be improved

First record of Trichinella in Leopardus guigna (Carnivora, Felidae) and Galictis cuja (Carnivora, Mustelidae): new hosts for South America

If it is the first record of Trichinella larvae in a native South American mustelid, isn't it the first record also in Galictis cuja?

Introduction

Line 37 Trichinellosis is a worldwide disease…..
Line 39-42 The paragraph should be improved
Trichinella is only transmitted by the consumption of infected meat, by predation or carrion consumption. So, circulates among carnivore and omnivore vertebrates: reptiles, birds and mammals. In human, trichinellosis is mainly associated to cultural factors, especially the slaughter of domestic pigs without veterinary inspection and the consumption of raw or poorly cooked meat.
Line 40 the genus Trichinella has been recorded in birds and reptiles as well as mammals. In some parts of the text it is correct. Please review.
Line 45-48 The paragraph should be improved
Thus, the knowledge of the hosts and reservoirs is relevant to propose control measures. The domestic and wild Trichinella cycles are connected and fed back primarily by invasive rats and other synanthropic animals (Pozio, 2000).
Line 51-55 The paragraph should be improved. The fact that most of the hosts are mammals has already been reported and is repeated. Perhaps the content of the introduction could be better organized.
-how is it transmitted
-what are the species
-which are the hosts
-which species circulate in host groups.
Line 58
In Bolivia, Brazil and Ecuador antibodies have been detected, but not the parasite. This difference must be considered.
Antibodies against Trichinella spp. have been detected in Bolivia, Brazil and Ecuador.
Please the paragraph should be improved.
Line 61-62 It remains to mention who described T. pseudospiralis
Line 60-64 Please check the singular and plural. For example: "sea lions", there are several species of sea lions and there is only one record for South America in one specimen of one species.
Line 67 Trichinella spiralis is the sole only species that has been reported in Chile
Line 71-73 Thus, the aim of this study was to assess the presence of Trichinella sp. in carnivorous or omnivorous wild vertebrates from south-central Chile.

Materials & Methods
Was not necessary a work permit from the relevant government department necessary to lift dead animals from the roads? In case it is, please mention it.

Line 76-77 The study area includes three administrative regions of Chile: the Maule, Ñuble and Biobío Regions (Fig. 1).

Line 80 The coordinates in the manuscript are not necessary, as the graphic contains them and is clear.
Line 87-95
The species and the number of individuals analyzed is a result, it should go under the subtitle results. They could be presented in table format, without repeating the information in the manuscript.
Examined animals were: Birds-Strigiformes: Bubo magellanicus Gmelin, 1788 (n=2), Glaucidium nana King, 1828 (1), Strix rufipes King, 1828 (2), Tyto furcata Temminck, 1827 (5), Birds-Accipitriformes: Parabuteo unicinctus Temminck, 1824 (11), Cathartes aura Linnaeus, 1758 (1), Coragyps atratus Bechstein, 1793 (2); Birds-Pelecaniformis: Pelecanus thagus Molina, 1782 (1); Mammals-Artiodactyla: Grampus griseus Cuvier, 1812 (1); Mammals-Carnivora: Otaria flavescens Shaw, 1800 (1), Leopardus guigna Molina, 1782 (6), Galictis cuja Molina, 1782 (17), Lycalopex culpaeus Molina, 1782 (9); Mammals-Microbiotheria: Dromiciops bozinovici D’Elía, Hurtado and D’Anatro, 2016 (1); and Reptile-Squamata: Philodryas chamissonis Wiegmann, 1834 (1).

Results
Line 115 (güiña. 52 larvae per gram)
Line 116 (lesser grison. 0.3 larvae per gram)
How many grams were digested from each positive animal? Please include the answer. It is important to the discussion.
What year were the samples of these animals collected? Because sampling was from 2013 to 2020. Please include the answer. It is important to the discussion.

Discussion

Line 121-123 The English language should be improved
Line 135 The English language should be improved.
Similarly, other felids have been reported harboring Trichinella in Chile, as Puma
They should mention that the species in which there was a positive specimen are the species with the largest number of samples (Galictis cuja 17) and another is among the most sampled (Leopardus guigna 6). Which could indicate that the lack of positive results could be related to a low number of animals sampled. Could it be that the lack of positive results is due to the number of grams analyzed?
Have these species been registered as a reservoir for other parasites or other pathogens that could indicate an important role in the connection between the wild and domestic cycle?

Conclusions
Line 182-183 It is appropriate to mention the specific names in the conclusion. If it is the first record of Trichinella larvae in a native South American mustelid, isn't it the first record also in Galictis cuja?

Make corrections also in the abstract.

Reviewer 2 ·

Basic reporting

The manuscript needs some attention regarding the writing style and I have made a few edits to the article and added a few comments/suggestions directly on the article and there are few places where further clarifications are required. I have written my comments/suggestions directly on the manuscript (attached). I recommend that the manuscript is further reviewed by a native English speaker to improve the language.

Experimental design

The design is acceptable as it is based on convenient samples based on the reasons given of challenges in accessing wildlife animals for this type of study.

Validity of the findings

The result could have been easily presented in a tabular form for clarity. I strongly suggest that the authors include photographs of the isolated larvae as well as the PCR product they got with the size indicated. Again, if possible, the best and non-equivocal way of establishing the species as Trichinella spiralis is to send the PCR products for sequencing. I recommend that this is done if these are still available,

Additional comments

N/A

Annotated reviews are not available for download in order to protect the identity of reviewers who chose to remain anonymous.

---

## Round 0.2 · Minor Revisions

Dr. Landeata-Aqueveque,

Your manuscript “First record of Trichinella in Leopardus guigna (Carnivora, Felidae) and Galictis cuja (Carnivora, Mustelidae): New hosts in Chile” requires minor revisions before it can be accepted. Although the writing is improved in this version of the manuscript, there are several areas where it needs improvement. The reviewers and I have made recommendations on improving those sections. My comments are in the attached .pdf document. Have a great day and I look forward to your revisions.

·

Basic reporting

Comments made in the first review.

Experimental design

Comments made in the first review.

Validity of the findings

Comments made in the first review.

Additional comments

Introduction
Line 67-71: include Pecarí (Tayassu tajacu) :
Soria, C.; Mozo, G.; Camaño, C.; Saldaño, B.; López, E.; Malandrini, J.; Soria, J. (2010) Aislamiento de larvas de Trichinella spp. en Pecarí (Tayassu tajacu) de Icaño, Departamento La Paz, Catamarca. Revista Iberoamericana de Tecnología en Educación y Educación en tecnología, 2(1):153-163.

In table 1: the internal laboratory code is not required. Please include a column with the vulgar name and Bierd (B) Mammals (M) or Reptil (R). Highlight the two positive animals as in figure 2 (Infected animals are presented with the symbols “+” (Leopardus guigna) and “*” (Galictis cuja). I would sort the samples by date.

Reviewer 2 ·

Basic reporting

The authors have adequately addressed my previous comments. However, the manuscript still needs some professional editing in may areas. I have attempt to correct some parts but still there is lot to be done to improve the language throughout the manuscript.

Experimental design

Nothing to add.

Validity of the findings

Nothing to add.

Additional comments

Nothing to add.

Annotated reviews are not available for download in order to protect the identity of reviewers who chose to remain anonymous.

---

## Round 0.3 · Minor Revisions

I feel you have addressed the major concerns with the science of the manuscript. However, there are still some issues with the flow and writing style in the manuscript. I have rewritten some sections of the manuscript and made a few changes with the scientific and common names used. I also suggest the authors change the common name of the Magellanic horned owl to lesser horned owl. The changes are in track changes in the attached the PDF document.

---

## Round 0.4 · accepted · Accept

You have addressed all of the concerns.